# Clinical and Radiological Outcome of Disc Arthroplasty for the Treatment of Cervical Spondylotic Myelopathy

**DOI:** 10.3390/jpm13040592

**Published:** 2023-03-28

**Authors:** Peter Obid, Anastasia Rakow, Gernot Michael Lang, Wolfgang Marx, Thomas Niemeyer, Tamim Rahim

**Affiliations:** 1Department of Orthopaedics and Traumatology, Freiburg University Hospital, 79106 Freiburg, Germany; 2Center for Orthopaedics, Trauma Surgery and Rehabilitation Medicine, Greifswald University Hospital, 17475 Greifswald, Germany; 3Spine and Scoliosis Center, Asklepios Klinik Wiesbaden, 65197 Wiesbaden, Germany

**Keywords:** cervical myelopathy, cervical disc arthroplasty, cervical total disc replacement, adjacent segment disease, outcome, CP ESP^®^

## Abstract

Introduction: The aim of this study is to evaluate the clinical and radiological results of cervical disc arthroplasty (CDA) in patients with cervical spondylotic myelopathy (CSM) using the CP ESP^®^ disc prosthesis. Materials and Methods: Prospectively collected data of 56 patients with CSM have been analyzed. The mean age at surgery was 35.6 years (range: 25–43 years). The mean follow-up was 28.2 months (range: 13–42 months). The range of motion (ROM) of the index segments, as well as upper and lower adjacent segments, was measured before surgery and at final follow-up. The C2-C7 sagittal vertical axis (SVA), C2-C7 cervical lordosis (CL), and T1 slope minus cervical lordosis (T1s-CL) were analyzed as well. Pain intensity was measured preoperatively and during follow-up using an 11-point numeric rating scale (NRS). Modified Japanese Orthopaedic Association (mJOA) score was assessed preoperatively and during follow-up for the clinical assessment of myelopathy. Surgical and implant-associated complications were analyzed as well. Results: The NRS pain score improved from a mean of 7.4 (±1.1) preoperatively to a mean of 1.5 (±0.7) at last follow-up (*p* < 0.001). The mJOA score improved from a mean of 13.1 (±2.8) preoperatively to a mean of 14.8 (±2.3) at last follow-up (*p* < 0.001). The mean ROM of the index levels increased from 5.2° (±3.0) preoperatively to 7.3° (±3.2) at last follow-up (*p* < 0.05). Four patients developed heterotopic ossifications during follow-up. One patient developed permanent dysphonia. Conclusions: CDA showed good clinical and radiological outcome in this cohort of young patients. The motion of index segments could be preserved. CDA may be a viable treatment option in selected patients with CSM.

## 1. Introduction

Cervical spondylotic myelopathy (CSM) is a progressive disease caused by the compression of the spinal cord due to degenerative changes in the cervical spine. Etiology of the compression may be acute in the form of a disc prolapse or chronic in the form of progressive segment degeneration. Other causes of chronic compression may be ossification of the posterior longitudinal ligament or the ligamentum flavum. More than 50% of the older population show degenerative changes of the cervical spine, 10% of whom have symptoms of spinal cord compression or cervical radiculopathy [1,2]. CSM is the leading cause of spinal cord dysfunction in the older population [3]. However, clinical presentation and the progression of symptoms may vary greatly [4]. Consensus exists for the surgical treatment of moderate and severe CSM [5]. In cases of mild CSM, surgery may be superior to conservative treatment as well, considering a significant number of patients (20% to 60%) will deteriorate over time if treated nonoperatively [6,7,8]. Historically, operative treatment of CSM has been performed to halt the progression of spinal cord dysfunction and to prevent further functional impairment of affected patients. Recent evidence suggests that surgical intervention for CSM is associated with improvement in function and health-related quality of life (hrQOL) [6,9].

Anterior cervical decompression and fusion (ACDF) is most commonly performed as operative treatment for CSM if the pathology can be adequately addressed via an anterior-only approach. ACDF may be associated with progressive adjacent segment disease (ASD) by increasing intradiscal pressure and segmental motion at levels adjacent to fusion during normal range of motion [10]. Cervical disc arthroplasty (CDA) aims to preserve motion at the index level of surgery, thus preventing ASD and reducing the rate of revision surgeries. Although the ability of CDA to actually achieve these goals is still unclear, some randomized trials have found lower rates of reoperation in patients treated with CDA as compared to ACDF [11,12,13]. Few studies have investigated the outcome of CDA in the context of CSM [14,15]. Additionally, the indication for CDA in the context of CSM is discussed controversially, since segmental instability may be one driver for the progression of CSM [16]. The aim of this study is to analyze the clinical and radiological outcome after CDA for the treatment of CSM.

## 2. Materials and Methods

We retrospectively reviewed all patients who underwent decompression and CDA due to CSM between January 2016 and December 2019 at a single center. A total of 56 patients with complete one-year follow-up could be identified. Three patients were excluded due to incomplete follow-up. All patients showed clinical and neurophysiological signs of CSM with or without additional radiculopathy due to disc prolapse and/or spinal stenosis with spinal cord compression at the cervical spine. Preoperatively, all patients showed pathological motor and somatosensory evoked potentials (MEPs and SSEPs). Preoperative neurological findings are shown in Table 1. Pain intensity was measured preoperatively and during follow-up using an 11-point numeric rating scale (NRS). Modified Japanese Orthopaedic Association (mJOA) score was assessed preoperatively and during follow-up for the clinical assessment of myelopathy [5]. All patients received magnetic resonance imaging (MRI) and anteroposterior and lateral flexion/extension radiographs of the cervical spine preoperatively. Radiographs were routinely repeated postoperatively and after 12 and 24 months. Range of motion (ROM) of the index segment and adjacent segments as well as C2-C7 sagittal vertical axis (SVA), C2-C7 cervical lordosis (CL), and T1 slope minus cervical lordosis (T1s-CL) were measured preoperatively and at last follow-up. Surgical complications were analyzed as well. 

### Implant and Surgical Technique

The CP ESP^®^ disc prosthesis (FH Orthopedics, Mulhouse, France) was used in all patients. It is a one-piece deformable implant including a central core made of polycarbonate urethane (PCU) fixed to titanium endplates. The endplates have anchoring pegs to provide primary fixation and are covered by a textured titanium layer and hydroxyapatite to improve bony ingrowth. The PCU annulus is stabilized by supplementary inner pegs located on the internal surface of both metal endplates. The implant provides six full degrees of freedom about the three axes [17]. Figure 1, Figure 2 and Figure 3 show exemplary radiographs of a patient after implantation of the CP ESP^®^ disc prosthesis.

A standard anterolateral approach was used in all cases. Prior to incision, the level was marked using fluoroscopy. Discectomy and decompression were performed using a microscope. During surgery, the segment was distracted using screws and a retractor. The size of the disc prosthesis was chosen using a trial implant and fluoroscopy. Care was taken to maximize the footprint of the prosthesis, to restore an optimal height of the segment, and not to induce overdistraction of the segment. Our contraindications for CDA are summarized in Table 2.

## 3. Results

A total of 56 patients were included, 32 females (59.2%) and 24 males (40.8%). The mean age was 35.6 years (range: 25–43 years) at the time of surgery. The mean follow-up was 28.2 months (range: 13–42 months). The main etiology of CSM was herniated discs in 47 patients (83.9%) and degenerative spinal stenosis in 9 patients (16.1%). CDA was performed at two levels in 13 patients, and at one level in 43 patients. The affected levels were C3/4 (*n* = 9), C4/5 (*n* = 14), C5/6 (*n* = 29), and C6/7 (*n* = 17). The NRS pain score improved from a mean of 7.4 (±1.1) preoperatively to a mean of 1.5 (±0.7) at last follow-up (*p* < 0.001). The mJOA score improved from a mean of 13.1 (±2.8) preoperatively to a mean of 14.8 (±2.3) at last follow-up (*p* < 0.001). Table 3 provides an overview of patients with mild, moderate, and severe myelopathy preoperatively and at last follow-up. Table 4 shows the mean ROM of the index and adjacent levels, C2-C7 sagittal vertical axis (SVA), C2-C7 cervical lordosis (CL), and T1 slope minus cervical lordosis (T1s-CL) preoperatively and at last follow-up. 

Complications occurred in five patients. Four patients developed heterotopic ossifications (HO) during follow-up: Grade 2 in three cases and Grade 1 in one case. HOs were graded according to Mehren et al. [18]. One patient developed permanent dysphonia due to an injury of the recurrent laryngeal nerve. No implant-associated complications or adjacent segment degenerations occurred during follow-up. There were no revision surgeries within the follow-up period.

## 4. Discussion

ACDF is the traditional gold standard for the surgical treatment of cervical degenerative disc disease. The number of CDAs performed in recent years has increased because of the postulated ability of cervical disc prostheses to preserve the motion of the index level and to prevent adjacent level hypermobility and, thus, minimize the occurrence of ASD [19,20]. Despite promising results, few studies have analyzed the clinical outcome of CDA in patients with CSM [14,15,21]. Additionally, it is still controversially discussed whether CDA should be performed in the setting of CSM [22,23].

The literature is ambivalent regarding persistent neck pain after CDA versus ACDF. Sasso et al. found significantly greater improvement for neck pain after CDA compared to ACDF [11]. On the other hand, Tracey et al. found a significantly higher rate of persistent neck pain after CDA than after ACDF [24]. A recent meta-analysis by Gendreau et al. did not identify statistically significant differences for neck or arm pain after ACDF or CDA [25]. We did not measure NRS separately for arm and neck pain but found a statistically significant improvement for combined arm and neck NRS from a mean of 7.4 (±1.1) preoperatively to a mean of 1.5 (±0.7) at last follow-up (*p* < 0.001). In the case of a proper indication, CDA is a reliable procedure to improve neck and arm pain.

It is still debated if CSM should be addressed by CDA because spinal segmental instability may be a contributing cause for the development of myelopathy. In this study, segmental instability was defined as angulation >11° or translation >3 mm on flexion/extension radiographs [26]. Our results show that CDA is a reliable treatment option, especially for young patients with CSM, if contraindications for the implantation of a cervical disc prosthesis are respected. The mJOA significantly improved from a mean of 13.1 (±2.8) preoperatively to a mean of 14.8 (±2.3) at last follow-up (*p* < 0.001). Additionally, the number of patients with severe or moderate CSM decreased (Table 3).

The ROM of index and adjacent levels are shown in Table 4. In our cohort, ROM was not only preserved but improved after surgery. The ROM of upper and lower adjacent segments also increased after surgery. The improvement in ROM after surgery is most likely due to the alleviation of pain. Our results support the hypothesis that ROM can be preserved through CDA during a short observation period. The combination of preserved ROM and satisfactory clinical and functional results may fulfill the requirements of CDA for avoiding ASD in young patients. ROM after surgery may also depend on the type of prosthesis used. In a retrospective study by Chang et al., the ROM of the index levels was compared after the implantation of three different cervical disc prostheses: CP ESP, M6-C, and Mobi-C. The ROM was significantly greater after the implantation of Mobi-C compared to M6-C and CP ESP. However, the ROM was not compared to preoperative values [27]. 

The SVA did not change after CDA. The CL significantly increased, and T1s-CL significantly decreased after CDA compared to preoperative values. Although normative values for T1s-CL have not been established, recent studies imply a relationship similar to the relationship of PI–LL (Pelvic Incidence–Lumbar Lordosis) in the lumbar spine [28,29]. The short-term results indicate that the restoration of cervical sagittal alignment is feasible in selected cases using CDA if there is no fixed deformity.

Our cohort of patients is relatively young (mean age: 35.6 years; range: 25–43 years). Usually, patients with CSM are older, and the pathogenesis is more likely due to advanced segmental degeneration with or without an additional herniated disc. In these cases, indication for CDA may be less likely. Nevertheless, our results show that in selected patients, CDA may be a viable option for the treatment of CSM.

### Limitations

The mean follow-up period is only 28.2 months (range: 13–42 months), which may be a reason for the absence of ASD in this study. Additionally, our study does not include a control group. Whether or not CDA is truly able to reduce revision and/or adjacent segment surgeries compared to ACDF in the long term must be evaluated through prospective randomized trials with a long follow-up period.

## Figures and Tables

**Figure 1 jpm-13-00592-f001:**
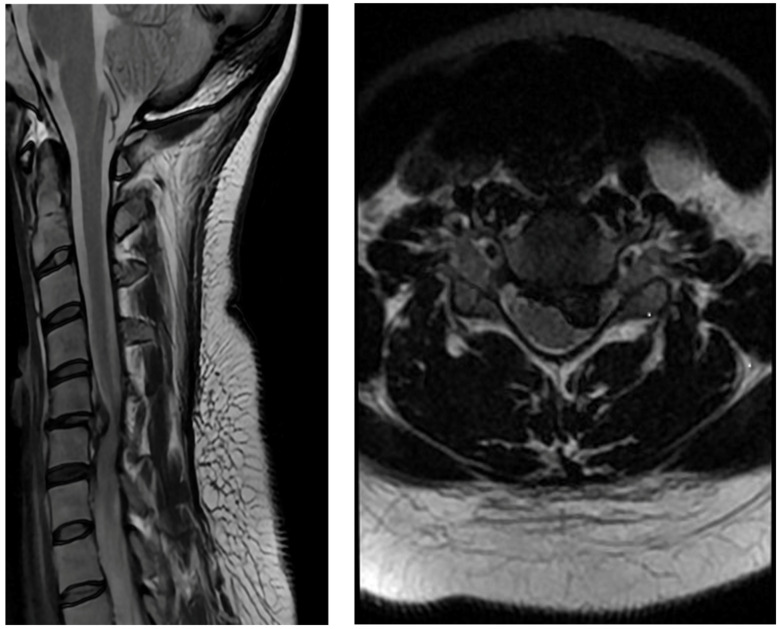
Preoperative MRI scan showing a large, herniated disc with spinal cord compression.

**Figure 2 jpm-13-00592-f002:**
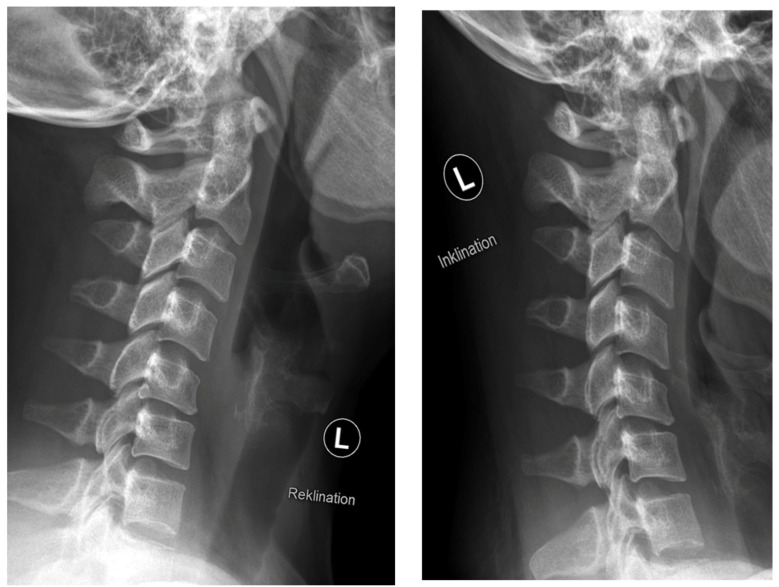
Preoperative radiographs in flexion (“Inklination”) and extension (“Reklination”).

**Figure 3 jpm-13-00592-f003:**
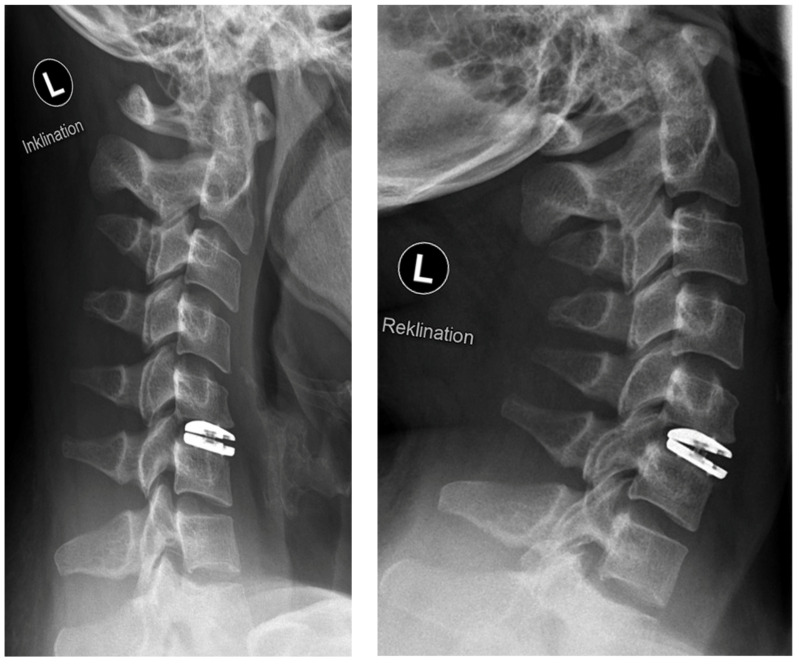
Postoperative radiographs in flexion (“Inklination”) and extension (“Reklination”).

**Table 1 jpm-13-00592-t001:** Preoperative neurological findings; cohort size: *n* = 56 patients.

Neurological Findings	Number of Patients (*n*, %)
Sensory disorders	*n* = 56, 100%
Radiculopathy	*n* = 29, 51.8%
Paresis	*n* = 24, 42.8%
Increased reflexes	*n* = 23, 41.1%
Lhermitte’s sign	*n* = 21, 37.5%
Gait ataxia	*n* = 20, 35.7%
Pathologic reflexes	*n* = 19, 33.9%
Clonus of lower extremities	*n* = 11, 19.6%
Paraspastic	*n* = 9, 16.1%
Atrophy of hand muscles	*n* = 2, 3.6%

**Table 2 jpm-13-00592-t002:** Contraindications for cervical disc arthroplasty.

Osteoporosis (*t*-score < −2.5)
Rheumatoid arthritis
Ankylosing spondylitis, diffuse idiopathic skeletal hyperostosis
Bridging osteophytes or absence of movement on flexion/extension radiographs
Segmental instability
Severe loss of disc height (>50%)
Previous trauma or surgery of the index segment

**Table 3 jpm-13-00592-t003:** Number (*n*) of patients/percentage (%) of the study cohort with mild, moderate, and severe myelopathy as defined by mJOA score preoperatively and at last follow-up.

mJOA Score	Number of Patients Preoperatively	Number of Patients at Last Follow-Up
15–17 (mild myelopathy)	*n* = 25 (44.7%)	*n* = 38 (67.8%)
12–14 (moderate myelopathy)	*n* = 18 (32.1%)	*n* = 13 (23.3%)
≤11 (severe myelopathy)	*n* = 13 (23.2%)	*n* = 5 (8.9%)

**Table 4 jpm-13-00592-t004:** Mean ROM of the index and adjacent levels, C2-C7 sagittal vertical axis (SVA), C2-C7 cervical lordosis (CL), and T1 slope minus cervical lordosis (T1s-CL) preoperatively and at last follow-up.

	Preoperatively	Last Follow-Up	*p* Value
Mean ROM [°] index levels	5.2 (±3.0)	7.3 (±3.2)	<0.05
Mean ROM [°] upper adjacent level	6.8 (±3.6)	7.7 (±4.1)	<0.05
Mean ROM [°] lower adjacent level	6.1 (±3.2)	8.2 (±4.3)	<0.05
Mean SVA [mm]	15.6 (±7.6)	15.7 (±6.1)	>0.05
Mean CL [°]	7.9 (±9.3)	11.5 (±7.3)	<0.05
Mean T1s-CL [°]	14.8 (±8.6)	8.8 (±11.2)	<0.05

## Data Availability

All data analyzed during the current study are available from the corresponding author on reasonable request.

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
