# Peer review of "Clinical and Radiological Outcome of Disc Arthroplasty for the Treatment of Cervical Spondylotic Myelopathy"

_jpm, 2023, doi:10.3390/jpm13040592_

Round 1
Reviewer 1 Report
Are there degrees of normal ROM in people without cervical pathology?
How did you measure cervical ROM?
What was the objective of the X-rays? You didn't measure any angle on them?
What is the Modified Japanese Orthopaedic Association? What is the minimum and maximum score. What does it measure?
You have measured pain intensity using the Numeric Rating Scale, however, I don't see anything about it in the results.
Is there an association between pain and ROM improvement? Does pain improve with surgery?
Reviewer 2 Report
Dear Authors,
There are numerous substantive issues to address:
1. In Figure 1, preoperative MR and CT imaging should be included to show more details of cervical spondylotic myelopathy.
2. In Figure 3, the lower endplate of the CP ESP disc prosthesis is angled with the bone notch at the upper endplate of the vertebral body in the postoperative lateral radiographs,and it seems mismatch with the vertebral notch, which lead to a relative inadequate coverage of the disc endplate, and it will affecting primary fixation and bone ingrowth.
3. In the “Materials and Methods”and “Results” sections, imaging evaluation indicators should included range of motion (ROM), sagittal lordosis angle, intervertebral disc height (IDH), and prosthesis displacement, subsidence, loosening, locking/fusion, and heterotopic ossification (HO), adjacent segment degeneration (ASD), and other complications. IDH is closely related to disc degeneration, ROM and sagittal curvature. Cervical curvature is an important imaging parameter after CDA. It has been reported that maintaining or increasing cervical curvature can improve clinical efficacy.
4. In the “Discussion” section, I suggest to include a chapter on literature summary and comparison with other types of CDA implants (like Bryan, Prestige, ProDisc C, Mobic C et al) and discussion what would you do differently or how to improve the clinical and radiological outcome? As cervical spondylotic myelopathy is still considered as a relative contraindication of CDA.
5. A more problematic issue is that the relatively short follow-up time in this series. The preliminary clinical and radiological results of CP ESP cervical disk prosthesis with 2-year follow-up have been reported in 2016 Eur J Orthop Surg Traumatol (PMID: 26341803, DOI 10.1007/s00590-015-1695-1).
Round 2
Reviewer 2 Report
I still recommend replacing a more perfect postoperative radiographs of the exemplary case.
Author Response
We would like to thank the reviewer again for his time and the constructive feedback.
1) I still recommend replacing a more perfect postoperative radiographs of the exemplary case.
- We have exchanged the radiographs.